# Large Angiomyxoma of the Umbilical Cord-Uncomplicated Rupture of Tumor Membranes at 32 Weeks of Gestation

**DOI:** 10.3390/diagnostics12061339

**Published:** 2022-05-28

**Authors:** Ija Lisovaja, Ivanda Franckevica, Natalija Vedmedovska

**Affiliations:** 1Department of Obstetrics and Gynaecology, and Department of Pathology, Riga Stradins University, Dzirciema 16, LV-1007 Riga, Latvia; ivanda.franckevica@gmail.com (I.F.); natalyved@googlemail.com (N.V.); 2Department of Pathology, Children’s Clinical University Hospital, LV-1004 Riga, Latvia; 3Fetal Medicine Unit, Riga Maternity Hospital, Miera 45, LV-1013 Riga, Latvia

**Keywords:** angiomyxoma, hemangioma, umbilical cord, prenatal diagnostic

## Abstract

Background: When a tumor of the umbilical cord is prenatally visualized, it is possible to propose the diagnosis depending on the sonographic appearance of the tumor. Angiomyxoma of the umbilical cord appears as a complex solid-cystic mass that is made of angiomatous component and myxoid stroma. When the tumor is diagnosed, serial ultrasound and doppler examinations are used to monitor the tumor’s size and the overall fetal well-being including doppler investigations and fetal growth. Angiomyxomas are not associated with fetal chromosomal pathologies. The cases of intrauterine rupture and fetal death was described in the literature. Case presentation: A 28 years-old pregnant woman was referred to our clinic for second opinion because of visualized umbilical cord tumor during second trimester ultrasound screening. The tumor gradually increased in size until 34th week of gestation, when the rupture of the cystic component was observed. The fetal doppler studies was normal during the course of pregnancy, we observed decreased AC and decreased estimated fetal weight. At the gestational age of the 37 weeks the labor was induced and heathy male infant was born. Conclusions: For the first time to our knowledge, we demonstrate the case of uncomplicated rupture of the cystic component of the angiomyxoma that lead to the possibility to manage the pregnancy conservatively without any compromise of the fetus.

## 1. Introduction

Angiomyxoma of the umbilical cord is a rare tumor arising from endothelial cells of umbilical vessels that can be detected prenatally as mixed solid-cystic masses [1]. To our knowledge, there are about 30 cases described and only few fetuses was delivered vaginally.

The formation of the umbilical cord occurs between the 6th and 8th week of gestation by the approximation of the omphalomesenteric duct/yolk sac and the allantoic duct with the body stalk of the developing embryo [2]. The umbilical vessels-two arteries and one vein develops from the allantoic vessels. These vessels support connective tissue that is called the Wharton’s jelly. Angiomyxoma consists of an angiomatous nodule, ranging in size from 0.2 to 18 cm, generally encompassed by edema and myxomatous degeneration of Wharton’s jelly [2]. The edema is due to high permeability of angiomatous vessels that are placed in a soft, gelatinous tissue [3].

Clinically, the most frequent presentation is that of an isolated umbilical cord anomaly without other fetal malformations, offering amniocentesis on a routine basis is not warranted from the available data [4]. Cases of umbilical cord angiomyxomas seen in association with cutaneous or visceral hemangiomas, i.e., port wine skin lesions, and single umbilical artery have been reported. Some cases of multiple lesions of the newborn were described with multiple hemangiomas in the liver, intestines, skin, and brain [5].

Impaired umbilical circulation with stenosis of umbilical vein and arteries is considered as the predisposing factor for fetal demise and fetal intrauterine growth restriction [6,7].

Sometimes polyhydramnios and hydrops fetalis can be present [8,9]. In one case, there was pulmonary hypoplasia in the newborn due to extensive accumulation of serous fluid in the fetal tissues and the body cavities [10].

## 2. Case

A 28-years-old woman, gravida 2, para 1, was referred to Riga maternity hospital Fetal medicine unit for the second opinion at 24 weeks of gestation. Woman had previously vaginal delivery, complicated by manual placental ablation. She had an otherwise uneventful medical history and denied smoking, use of alcohol, illicit drugs or medications. Detailed ultrasound examination showed a single anatomically normal fetus, with appropriate for gestational age biometric measurements. During the examination the multilocular solid tumor measuring 9.6 × 7.8 × 6.7 cm (265.993 cm^3^) at the middle portion of the umbilical cord was observed (Figure 1). The differential diagnosis comprised umbilical cord teratoma and angiomyxoma.

The size of the solid component was 3.4 × 2.3 cm and did not change during the pregnancy. The umbilical arteries and vein were running around and through, extra vessels were not observed in the tumor (Figure 2a). At the follow-up, the mass had rapidly enlarged in the size, from 285.32 cm^3^ at 26 weeks to 1929.36 cm^3^ at 32 weeks (Figure 2b). As well the reduction of fetal growth was observed from 30th week of gestation (Figure 3b).

At 34 weeks of gestation reduction in the size of tumor’s cystic component with freely floating superficial membranes was observed (Figure 3a). That resembles the process of rupture of the membranes without any signs of bleeding. Normal ACM PSV (49.4 cm/s) was recorded during examination of the fetus. At that time the umbilical mass decreased in size enormously and measured 120.45 cm^3^. 

At 37 + 2 weeks the size of the tumor reduced to the 51.46 cm^3^. At this time fetal estimated weight was at 10th percentile, so it was decided to induce the labor 4 days later. A male infant was delivered vaginally 2750 g and 49 cm. The Apgar score of the baby was 8 and 9 at 1 and 5 min, respectively.

The placenta was delivered uneventfully. The placenta weighted 436 g and measured 18 × 14 cm. The cystic lesion of the cord was 5 cm in diameter (Figure 4e), located 22 cm away from umbilical cords insertion point. On microscopic examination, small capillary vessels were CD31 and CD34 antigen positive (Figure 4a,b). Around the vessels degenerated myxoid stroma was observed (Figure 4c,d). The umbilical cord coiling index was higher—0.4/cm (normally 0.1–0.3/cm). The histological conclusion was the angiomyxoma of the umbilical cord.

## 3. Discussion

The latest development of high-resolution ultrasound machines and 3D technology yields diagnostic accuracy of umbilical cord anomalies such as true knots, cord aneurysms, cysts, hematomas, excessive/ absent coiling, true cord tumors etc. [11,12,13]. Volume acquisition of the vessels by static 3D in combination with color Doppler can be used to assess angioarchitecture and differentiate vascular malformations applying different displays, including glass-body mode, when blood flow can be visualized together with the surrounding structures [14]. In our case the tumor was detected by 2D mode and additional application of the high definition (HD) flow Doppler allowed to delineate the course of blood vessels, their coiling and blood flow within the tumor. High resolution images may provide auxiliary information to the real 2D ultrasound, therefore facilitate management, patient consultation and even prognostication.

In our case the angiomyxoma was progressively growing from 24 weeks until 32 weeks. That have leaded to slowly decrease in estimated fetal weight from 50% at 24 weeks to 10% at 37 week of gestation and reduced abdominal circumference (AC) from 45% at 24 weeks to 1.1% at 34 weeks and then slightly increase in AC to 4.1% at 37 weeks. The explanation of this can be rupture and decrease in size of the angiomyxoma that have promoted better flow in umbilical vessels.

Serial ultrasound and Doppler examinations are used to monitor size of the angiomyxoma, look for evidence of cardiac compromise and evaluate overall fetal well-being [4,15,16]. Mapping of the fetal vessels through the mass has been advised to assess the potential risk of rupture of the cystic portion of the mass leading to tearing of the umbilical vein, and fetal hemorrhage [17]. In our case the rupture of cystic component occurred without any fetal compromise and even facilitated the AC growth. To our knowledge it is the only described case in literature of intrauterine rupture with benign outcome. 

In most cases of large angiomyxomas Cesarean section has been the reported mode of delivery [4,18]. Cases of ultrasound-guided in utero decompression following vaginal delivery and cases of spontaneous delivery were described. Concerns regarding intrapartum management without cystic decompression include possible risks of dystocia or sudden cystic rupture affecting blood flow through the cord. Controlled aspiration with ultrasonographic guidance may be used to decrease these potential risk factors [19]. Generally, the mode of delivery depends on the size and location of the tumor. In our case the reduction in size increased the possibility of vaginal delivery. 

## 4. Conclusions

Angiomyxoma is a rare umbilical cord tumor that can cause complications such as fetal intrauterine growth restriction, polyhydramnios, non-immune hydrops fetalis and intrauterine fetal death.There are no clinical guidelines how to manage these pregnancies. Every case is discussed individually, considering gestational age, coexisting pathologies, fetal growth and maternal state.To our knowledge this case demonstrates for the first-time uncomplicated rupture of the cystic component of the angiomyxoma.

## Figures and Tables

**Figure 1 diagnostics-12-01339-f001:**
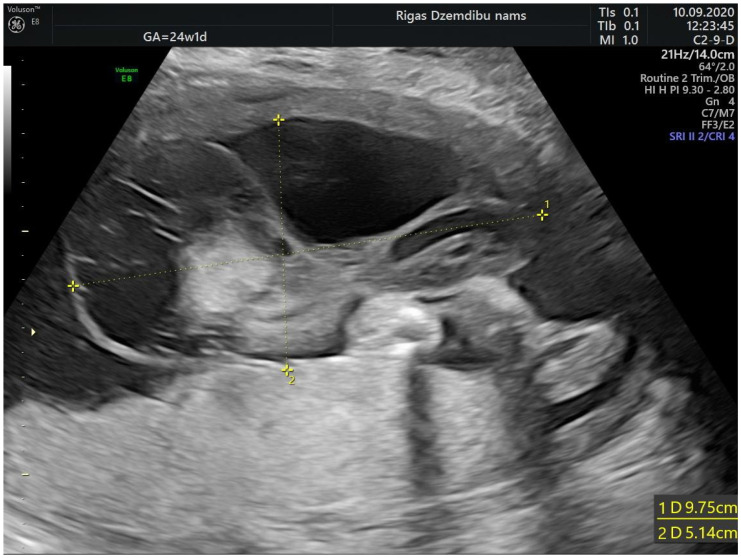
Tumor appearance at 24 + 1 weeks.

**Figure 2 diagnostics-12-01339-f002:**
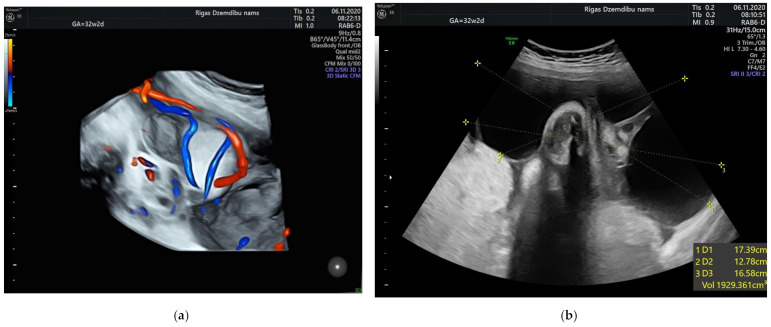
(**a**) Umbilical vessels going through the tumor at 32 + 2 weeks by 3D static HD flow (glass mode) imaging; (**b**) Tumor appearance at 32 + 2 weeks.

**Figure 3 diagnostics-12-01339-f003:**
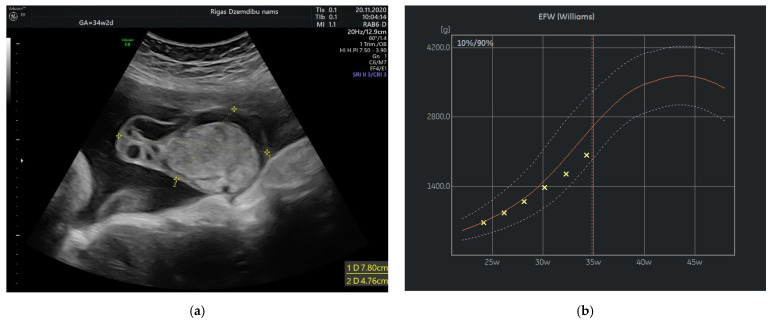
(**a**) Tumor appearance at 34 + 2 weeks; (**b**) Growth chart, estimated fetal weight by weeks.

**Figure 4 diagnostics-12-01339-f004:**
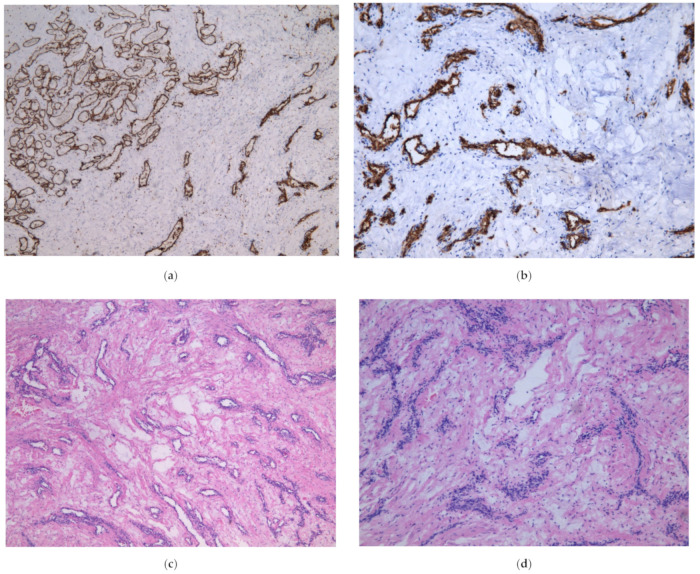
(**a**) Microscopic appearance of the tumor, CD31 in thin wall of the vessels, ×5; (**b**) Microscopic appearance of the tumor, CD34 in thin wall of the vessels, ×10; (**c**) Microscopic appearance of the tumor, small anastomosing vascular structures surrounded by myxoid degeneration, ×5; (**d**) Microscopic appearance of the tumor, obliterated small vessels in myxoid degeneration, ×10; (**e**) Macroscopic appearance of the tumor.

## Data Availability

Not applicable.

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
