# Peer review of "Large Angiomyxoma of the Umbilical Cord-Uncomplicated Rupture of Tumor Membranes at 32 Weeks of Gestation"

_diagnostics, 2022, doi:10.3390/diagnostics12061339_

Round 1

Reviewer 1 Report

I read with great interest the manuscript, which falls within the aim of this Journal. In my honest opinion, the topic is interesting enough to attract the readers’ attention. Nevertheless, authors should clarify some points and improve the discussion, as suggested below.

Authors should consider the following recommendations:

  • Manuscript should be further revised in order to correct some typos and improve style.
  • Authors should add further elements to discuss the potential use of 3D color Doppler ultrasound imaging for the diagnosis of umbilical cord diseases (authors may refer to: PMID: 27040420; PMID: 33947354).

Author Response

Dear Reviewer,

Thank you for taking your time to review our work. 

I have improved some typos. As well I have added about 3D in discussion. 

Please find the improvements in the revised paper.

Best regards,

Ija Lisovaja

Reviewer 2 Report

Very well documented case. The idea of long term observation and natural delivery is controvesial, but in carefull and profesional supervision is posssible.

Interesting.

Author Response

Dear Reviewer,

Thank you for taking the time to review our work! I am very happy to hear a good review from you.

Best regards,

Ija Lisovaja

Reviewer 3 Report

The manuscript presents an interesting and rare case report with an additional up-to-date review of the currently available literature in the discussion section. It seems well written. However, there are some issues that need to be addressed. 

More specifically, my main concern is about the lack of introduction/ background section with a respective reference on the aim of the study. I would recommend that the authors should add this section to their manuscript.

Additionally in line 60 at the end of presentation of the case report it is recommended that the reference on the microscopic examination  the section should be completed by adding the diagnosis according to the histology.

Author Response

Dear Reviewer,

Thank you for your time to review our work.

I have taken into account your advise and made corrections to my article.

Please find corrections in the revised paper.

Best Regards,

Ija Lisovaja